# Nurses’ Contributions in Rural Family Medicine Education: A Mixed-Method Approach

**DOI:** 10.3390/ijerph19053090

**Published:** 2022-03-06

**Authors:** Ryuichi Ohta, Satoko Maejma, Chiaki Sano

**Affiliations:** 1Community Care, Unnan City Hospital, 699-1221 96-1 Iida, Daito-cho, Unnan 699-1221, Japan; 2Department of Nursing, Unnan City Hospital, 699-1221 96-1 Iida, Daito-cho, Unnan 699-1221, Japan; satoko.m.lynette@gmail.com; 3Department of Community Medicine Management, Faculty of Medicine, Shimane University, 89-1 Enya cho, Izumo 693-8501, Japan; sanochi@med.shimane-u.ac.jp

**Keywords:** family medicine education, family medicine physicians, nurses, residents, rural hospitals, patient care, collaboration, difficulties, professionalism

## Abstract

Family medicine residents frequently collaborate with nurses regarding clinical decisions and treatments, which contributes to their education. In rural areas, these residents experience a wider scope of practice by collaborating with nurses. However, nurses’ contributions to rural family medicine education have not been clarified. This study measured the contributions of 88 rural community hospital nurses to family medicine education using a quantitative questionnaire and interviews. The interviews were recorded, transcribed verbatim, and analyzed using the grounded theory approach. Nurses’ average clinical experience was 20.16 years. Nurses’ contributions to the roles of teacher and provider of emotional support were statistically lower among participants working in acute care wards than those working in chronic care wards (*p* = 0.024 and 0.047, respectively). The qualitative analysis indicated that rural nurses’ contributions to family medicine education focused on professionalism, interprofessional collaboration, and respect for nurses’ working culture and competence. Additionally, nurses struggled to educate medical residents amid their busy routine; this education should be supported by other professionals. Rural family medicine education should incorporate clinical nurses as educators for professionalism and interprofessional collaboration and as facilitators of residents’ transition to new workplaces. Subsequently, other professionals should be more actively involved in improving education quality.

## 1. Introduction

Family medicine education involves various clinical experiences that broaden the scope of practice for family medicine residents. In their experience, nurses are one of their most frequent collaborators [1,2]. The scope of practice refers to the range of healthcare issues medical professionals can treat [3]. Upon graduating from residency, family physicians are expected to treat various healthcare issues related to patients’ biopsychosocial problems and frequently collaborate with nurses making clinical decisions and providing treatment [4]. 

In family medicine education, collaboration with nurses is vital for the development of family physicians’ competence. Nurses observe residents frequently in clinical situations, assess their skills, and provide feedback to improve their knowledge, skills, and attitudes [5,6,7]. For the effective provision of family medicine education, the involvement of nurses can be critical, leading to a better quality of care for patients [8,9].

Residents may not be able to perform proper clinical reasoning and decision-making in clinical situations while under pressure despite having the proper medical knowledge [10,11,12]. In such situations, nurses’ observations and feedback to residents and senior doctors can improve the quality of family medicine education. Through collaboration with senior doctors and nurses, patients can manage their problems more smoothly [13,14].

Previous studies have shown that nurses can act as practitioners of patient safety and educators of medical residents [15,16]. Their observations and feedback to medical residents and senior doctors can make patients safer and lead to more effective patient care [15,17,18,19]. The level of nurses’ contributions to physicians’ education can differ depending on clinical situations, such as medical resources, the number of senior doctors, and the specialty of education.

Rural family medicine residents experience a wide scope of practice owing to their demand in rural hospitals; therefore, nurses’ support and feedback can be more valuable in rural family medicine education [20]. Rural family medicine education may involve various conflicts due to systemic and cultural changes for medical residents, as they may have to change their working styles in adjusting to rural clinical situations [21,22,23]. In these processes, as the number of senior doctors is low, nurses may play critical roles in supporting residents’ conflicts [24]. As nurses frequently observe the residents, they could provide residents with various educational recommendations to improve their collaboration. Moreover, effective support and safe netting for patient care should be provided. Furthermore, in rural areas, there is a lack of physicians; therefore, interprofessional collaboration facilitates a good medical education in rural family medicine [25,26].

Currently, there is a lack of evidence regarding nurses’ contributions to rural family medicine education [15]. In addition, nurses’ difficulties in rural family medicine education have not been clarified. Therefore, the research question was “How do nurses contribute to rural family medicine education, and what difficulties do they experience?” By clarifying nurses’ contribution and difficulties in rural family medicine education, a concrete revision of such education can be executed, which may lead to better interprofessional collaboration in patient care education. Therefore, this research aimed to clarify nurses’ contributions and difficulties in rural family medicine education using a mixed-method approach.

## 2. Materials and Methods

This mixed-method research was conducted to investigate nurses’ contributions and difficulties in relation to family medicine education in a rural hospital using questionnaires (quantitative method) and interviews (qualitative method). Ethnography and interviews were conducted to clarify how nurses perceive their role in rural family medicine education, including the difficulties and the support required. The study duration was from 1 April to 31 December 2021. The researchers were participatory observers; they informed the residents and discussed the study application with the nurses. Additionally, a questionnaire regarding nurses’ ideas of their roles in family medicine residents’ education was provided to the participants to investigate their ideas quantitatively [15].

### 2.1. Setting

Unnan City is one of the smallest and most remote cities in Japan and is located southeast of an administrative unit in a rural setting. In 2020, the total population of the city was 37,638 (18,145 males and 19,492 females), and 39% were aged over 65 years; this statistic is expected to reach 50% by 2050. This city has 16 clinics, 12 home care stations, 3 visiting nurse stations, and only 1 public hospital. 

At the time of the study, the Unnan City Hospital had 281 care beds:160 acute care, 43 comprehensive care, 30 rehabilitation, and 48 chronic care beds. The nurse-to-patient ratios were 1:10 for acute care, 1:13 for comprehensive care, 1:15 for rehabilitation, and 1:25 for chronic care. The hospital had 27 physicians, 197 nurses, 7 pharmacists, 15 clinical technicians, 37 therapists, 4 nutritionists, and 34 clerks [27].

### 2.2. Educational Curriculum of Family Medicine Education in Unnan City Hospital

The educational curriculum is based on the Japanese Primary Care Association’s Board of Family Medicine, which was developed according to the World Standard of Education of Family Medicine [28]. In this curriculum, residents experience various clinical situations with their patients. In their first year, residents worked at a community hospital (Unnan City Hospital) for one year and treated typical diseases in both inpatient and outpatient situations. Additionally, they worked at a rural clinic for 6 months to learn home care and community-oriented primary care. To broaden their scope of practice regarding internal medicine, pediatrics, and emergency medicine, they worked at a general hospital. Each clinical setting included a medical teacher. 

Residents learned content through cognitive apprenticeship, legitimate peripheral participation, and continuous reflection with medical teachers and students [29]. The formative and summative assessments of the learners were accomplished using Mini-CEX, multiple-source feedback, and portfolios. After 3 years of training, the residents undergo a national examination in family medicine and obtain a family physician’s certificate [20,21]. In the first year of the training, which began on 1 April medical residents collaborated with various medical professionals at the community hospital. This curriculum can be utilized to educate a maximum of three residents simultaneously. One resident in 2018 and 2019 and three in 2020 and 2021 engaged in the curriculum.

### 2.3. Participants

The participants were registered nurses working in a rural community hospital. They were informed of the research purpose and agreed to participate. They were chosen from all hospital wards. The registered nurses who participated in this study had experience with physicians and nurses in the hospital. In addition, the qualitative interviews involved nurses who were charged with the administration of each ward. Each ward had two or three nurses with administrative roles, and all of them were requested to participate in this research and consented to participation. Overall, 88 nurses completed this distributed questionnaire, and of these, 20 nurses with administrative roles were interviewed based on the results of their questionnaires.

### 2.4. Data Collection

#### 2.4.1. Questionnaire

A questionnaire was provided to the participants regarding their roles in family medicine residents’ education. Based on a previous study, seven items were constructed with respect to the concepts of previous research: nurses as teachers, guardians of patient well-being, providers of emotional support, providers of general support, expert advisors, navigators, and team players [15]. The seven items were as follows: nurses need to provide educational support to medical residents to protect the safety of patients (Item 1: guardians of patient well-being); nurses need to convey the background and concrete information of patients and their families to support the medical care of residents (Item 2: navigators); nurses need to teach how nurses work in wards and how to prepare medical equipment so that residents can work smoothly in wards (Item 3: providers of general support); nurses need to support the medical care of residents to grow their personality and ability (Item 4: nurses as teachers); nurses need to play a supporting role in the emotional changes of residents (Item 5: providers of emotional support); nurses need to provide residents with knowledge regarding patient care as nursing specialists (Item 6: expert advisors); and nurses need to help residents make appropriate decisions in patient care (Item 7: team players). Each item was answered on a five-point Likert scale ranging from strongly agree (five) to strongly disagree (one). In addition, the gender, clinical experience, workplace, and educational background of the participants were collected.

#### 2.4.2. Ethnography and Semi-Structured Interviews

The first author performed ethnography and semi-structured interviews with the participants. This researcher’s specialties were family medicine, medical education, and public health. The researcher worked in all hospital wards, observed the interaction between residents and nurses in each ward, and took field notes during this process. During the observation period, the researcher interviewed the nurses. The interviews were performed based on the questionnaire results. The interview guide included three questions.

The first question was “How do you feel about the current family medicine education in community hospitals?”

-The follow-up questions focused on the accomplishment of their education of family medicine residents.

The second question was “What do you think you can do in family medicine education at a community hospital?”

-The follow-up questions focused on how the nurses educated family medicine residents as per the positive quantitative results for the questionnaire items.

The third question was “How do you think your role impedes family medicine education at a community hospital?”

-The follow-up questions focused on how the nurses educated family medicine residents as per the negative quantitative results for the questionnaire items.

Each interview lasted approximately 30 min and was recorded and transcribed verbatim. The transcript was shared with the interviewees to confirm the credibility of the content. 

### 2.5. Data Analysis

Quantitative data were analyzed using Student’s t-test and a Chi-square test for the background data. The results of each question regarding nurses’ roles in family medicine education were compared between the characteristics of the wards in which the participants worked: acute or chronic care using Student’s t-test. Regarding qualitative data, the grounded theory approach was used to clarify nurses’ contributions and difficulties in regard to rural family medicine education in rural community hospitals. The first and second authors carefully and thoughtfully read the field notes and transcriptions. After reading them in depth, the third author coded the contents and developed codebooks based on repeated reading of the research materials as the initial coding [30]. This study used process and concept coding [31]. The first author also coded the materials and discussed the coding and codebooks with the third author for coding refinement. In the second coding, the first and third authors induced, merged, deleted, or refined the concepts and themes by going back and forth between the research materials and initial coding [30]. This process was performed through constant discussion until mutual agreement was reached and repeated until no new codes or concepts appeared, indicating saturation. For member checking, analysis was provided to all participants, whose feedback was then included in the final revision of themes and concepts. Eventually, no new themes emerged during member checking, indicating saturation. Finally, the theory was discussed by two authors who ultimately reached an agreement on the final theory.

### 2.6. Ethical Consideration

Before this study, the participants were informed that the collected data would only be used for research purposes. They were also informed of the research aims, how the data would be disclosed, and how their personal information would be protected. The participants then provided written informed consent. This study was approved by the Unnan Hospital Clinical Ethics Committee (approval code: 20210022).

## 3. Results

### 3.1. Results of the Questionnaire on Nurses’ Roles in Family Medicine Education

The nurses’ average clinical experience was 20.16 years (standard deviation [SD] = 8.86), and most participants graduated from specialized nursing schools. All of the participants were women. Regarding the questionnaire, the scores for items on “nurses as teachers” and “providers of emotional support” were statistically lower among the participants working in acute care wards than those working in chronic care wards (*p* = 0.024 and 0.047, respectively). The other items regarding nurses’ roles—“guardians of patient wellbeing”, “navigators”, “providers of general support”, “expert advisors”, and “team players”—had higher scores than the roles of “nurses as teachers” and “providers of emotional support” but did not report any significant statistical differences (Table 1).

### 3.2. Results of the Qualitative Analysis Regarding the Nurses’ Roles in Family Medicine Education

A total of 34 pages of fieldnotes were composed. A total of 20 nurses were interviewed (9 from acute care wards and 11 from chronic care wards). There were eight nursing directors and 12 semi-directors. Three themes and eight concepts were extracted using the grounded theory approach. The themes were nurturing professionalism, driving interprofessional collaboration, and respect for the environment and nurses’ competence (Table 2). Each theme was explained based on the relevant concepts and related quotations.

### 3.3. Nurturing Professionalism

The nurses interacted with the family medicine residents and realized the residents should improve their quality of professionalism while caring for patients as medical doctors. They attempted to discuss these behaviors with the residents as authentic physicians. Residents tended to decide on various treatments and care based primarily on medical aspects. However, the nurses attempted to incorporate into residents’ decision-making respect for patients’ backgrounds to facilitate effective care because various ethical decisions are made in geriatric medicine. When the residents struggled with ethical decision-making, the nurses conversed with the residents to support their discussions with patients and their families to effectively address these issues. Consistent with our quantitative results, nurses functioned as guardians of patient well-being, navigators, and expert advisors.

#### 3.3.1. Responsibility as a Physician

The nurses realized that residents needed to modify their attitudes toward patients, families, and other medical staff as professional physicians, which could be supported by the nurses. Family medicine residents were trained in various medical situations by their teachers but had no experience of authentic responsibility for patient care in previous situations. In the rural hospital, they had to determine patients’ treatment and care plans in outpatient and inpatient departments. In these processes, the residents’ ambiguous attitudes and vague decision-making confused the nurses. Nurses tried to modify residents’ attitudes toward medical care. They realized that before determining medical care, medical residents should nurture themselves as authentic physicians. 

Participants stated the following:
“*I understand that the medical residents learned a lot about medical issues. However, their attitudes toward patient care may not be authentic.*”(Participant 1, acute care ward)
“*Medical residents’ vague attitude toward patients is dangerous for patient care. They should recognize the responsibility of doctors as medical professionals.*”(Participant 5, chronic care ward)
“*The residents’ attitudes as professionals can be nurtured through the experiences and discussions with teachers and us. Therefore, I think that nurses play a role in improving residents’ professionalism.*”(Participant 2, acute care ward)

The nurses experienced difficulties in patient care due to the residents’ low quality of professionalism; however, they attempted to discuss the residents’ professionalism and nurture them through clinical experiences and collaboration with nurses.

#### 3.3.2. Respecting Patients’ Backgrounds

The nurses observed that the medical residents did not respect patients’ backgrounds and needed to include psychosocial aspects in their decision-making. Residents’ decision-making primarily focused on biomedical conditions and did not consider patients’ lives in their homes or nursing homes. The nurses knew that the medical residents learned the biopsychosocial model, in which family physicians understand patients from not only a biomedical perspective but also a psychosocial perspective to provide better care. However, the residents’ skills and attitudes needed to be enhanced through clinical experience and collaboration with nurses. 

Participants stated the following:
“*The medical residents did not understand the patients’ lives in their homes. They should respect the patients’ lives while respecting their quality of life at home. Their medical decisions may improve medical conditions, but they detach patients to discharge in their home. Their decisions should be supported by nurses considering various patient contexts*.”(Participant 12, chronic care wards)
“*I understand that the medical residents tried to understand patients’ conditions from various perspectives. Understanding patients in the context of their lives requires numerous experiences. Nurses try to support their learning and medical decisions with respect to the patient’s background. Medical residents can accept our suggestions and improve their skills and attitudes effectively.*”(Participant 3, acute care wards)

The nurses realized that they played a role in education regarding the biopsychosocial approach. Their support in resident education required humility and honesty from the medical residents.

#### 3.3.3. Enhancing Ethical Attitude

Through residents’ improvement as authentic physicians and respecting patients’ backgrounds, the nurses believed that medical residents could learn about ethical attitudes in the treatment of older and frail patients through discussion with nurses. Medical residents are exposed to various ethical issues when caring for older patients. The residents struggled with ethical decision-making because of their lack of experience. The nurses realized that they could effectively support the residents through dialogue regarding patients’ quality of life. 

Participants stated the following:
“*Ethical decisions are complicated and challenging. Medical residents often struggle to consider their patients’ conditions, such as discussions about life extension, gastrostomy, and the possibility of home care*.”(Participant 20, chronic care ward)
“*The medical residents lack the experiences of decision making about ethical issues. Thus, experienced nurses support their decision making and inform them about the patient’s context and their family’s ideas for their lives*.”(Participant 7, acute care ward)

The nurses respected patients’ lives from various perspectives and supported medical residents in making ethical decisions by providing important information when they were struggling. Nurses’ contribution to modifying residents’ attitudes as authentic physicians and incorporating respect for patients’ backgrounds improved residents’ professionalism and enhanced their ethical consideration.

### 3.4. Driving Interprofessional Collaboration

Nurses emphasized the importance of collaboration. They believed that medical teachers, as well as nurses, should first give detailed feedback to the residents regarding effective collaboration. In clinical situations, the nurses educated residents regarding the collaboration among medical professionals. In addition, nurses considered that through various educational insights and experiences, the medical residents could realize the effectiveness of interprofessional collaboration in improving patient care. Thus, consistent with the quantitative results, nurses functioned as providers of general support and team players.

#### 3.4.1. Getting Feedback from Teachers and Nurses

Nurses realized that medical students could collaborate with other medical professionals regarding their mental conditions and discuss their collaborative ideas with medical teachers and nurses. Mutual understanding is essential for collaboration. The nurses hoped to educate residents regarding nurses’ roles in resident education through discussions with medical teachers. 

Participants stated the following:
“*I tried to educate the medical residents. I hope to learn how to improve my role in their education. To get feedback on the education, medical teachers and residents should discuss education with and provide feedback to the nurses*.”(Participant 5, chronic care ward)
“*Collaboration between the teachers and the residents is essential. As one of the teachers, I want to improve my role in education and try to give and obtain feedback from residents and medical teachers*.”(Participant 11, acute care ward)

Nurses wanted mutual feedback on residents’ education to improve their role in the education. For improvement, medical teachers and nurses need to provide feedback to the residents regarding interprofessional collaboration. Moreover, providing feedback on nurses’ contributions to resident education is required to improve education.

#### 3.4.2. Importance of Dialogue with Other Professionals

Nurses considered that medical residents could learn more about the importance of dialogue with other professionals. In practice, the residents had to collaborate with various professionals as care for older patients requires various types of professional care during and after admission. The collaborative process demanded that the residents engage in discussions with other professionals. During this process, nurses educated the residents regarding dialogue with other professionals. 

One of the participants stated, “*I had many discussions with the medical residents regarding patient care. Initially, they attempted to collaborate with other professionals. However, they tended to be persistent in their opinions without accepting nurses’ ideas. They should have made decisions based on dialogue with the nurses*” (Participant 13, chronic care ward).

Through frequent dialogue with nurses and other medical professionals, medical residents can collaborate with nurses effectively. The nurses realized that their facilitation and the residents’ acceptance of the nurses’ positive feedback improved their interprofessional collaboration. 

Participants stated the following:
“*The medical residents could improve their understanding and skills in collaboration with other medical professionals. Now, they are trying to respect the various ideas of other professionals*.”(Participant 6, acute care ward)
“*They could become honest in patients’ care through various clinical experiences. They could now accept various ideas*.”(Participant 10, chronic care ward)

#### 3.4.3. Quality Improvement of Care through Collaboration

For better patient care, the nurses considered that residents’ education should include teaching the effectiveness of interprofessional collaboration. The nurses made an effort to support the residents to facilitate a better quality of care for the medical team. 

One participant stated:
“*For a true understanding of the importance of interprofessional collaboration, residents should understand the effectiveness of interprofessional collaboration. Therefore, I have tried to advise medical residents to improve patient care by respecting other professional perspectives*.”(Participant 19, chronic care ward)

Understanding the effectiveness of interprofessional collaboration in clinical experience has improved residents’ understanding. The nurses observed the medical residents’ changing skills and attitudes toward interprofessional collaboration. 

One of the participants stated, “*I think that medical residents could realize the effectiveness of interprofessional collaboration. In the dialogue and discussion with the medical residents, I felt that they tried to obtain advice from other professionals*” (Participant 2, acute care ward).

### 3.5. Respect for the Environment and Nurses’ Competence

The nurses considered that medical residents should respect the culture of wards and nursing. As the medical residents came to a rural hospital from a tertiary hospital, their working environment changed drastically. The initial work of the residents did not match the ward environment and nursing culture. The nurses attempted to educate residents on their working styles. In addition, the nursing education of the residents was burdensome in regard to their work. Nurses believed that their competence and their educational burden should be respected by medical teachers and residents. As reported by the quantitative results, the nurses realized their educational role as nurses acting as teachers and providers of emotional support, but their working conditions led to inadequate support for medical residents.

#### 3.5.1. Understanding Working Environments and Culture

The nurses considered that medical residents had changed their working style in the hospital. Initially, residents’ working styles impinged on nurses’ usual jobs. In addition, the nurses believed that residents should understand how nurses think and work in each ward.

One of the participants stated, “*The medical residents tended to order various tests for patients with various timing, not respecting the nurses’ work. Emergency situations may require temporal testing for diagnosis and treatment, but in their situations, they should have ordered them by observing the nurses’ work and asking us whether the testing was possible*” (Participant 8, chronic care ward).

The nurses had their own culture regarding patient care and their respected order of work, such as ways of approaching patients. The dialogue between the medical residents and nurses as well as nurses’ education of residents related to their working styles and work culture stimulated the residents’ learning regarding the culture, which moderated their behaviors.

One participant stated, “*The dialogue with the medical residents was important. By having various conversations with the residents, they gradually understood our working styles and culture, especially approaches toward patients. I concretely educated some residents about the nurses’ working styles. The residents tried to change their attitudes and timing in approaching the patients to avoid interrupting nursing care*” (Participant 18, chronic care ward).

#### 3.5.2. Working with Respect for the Nurses’ Competence

The nurses considered that the working structure for nurses affected their role in medical residents’ education. They had many routine tasks and could not support residents who were experiencing mental stress. The nurses felt that their work-related competence should be respected. 

One of the participants stated, “*Nurses’ routine work is tight. I feel that I should follow the residents’ feelings after dealing with the severe situations of their patients, but I could not support the medical residents dealing with difficult cases with social problems frequently because of the difficulties of working*” (Participant 9, chronic care ward).

In rural contexts, the workforce is limited to rural hospitals. The nurses hope that their educational role is respected; however, the follow-up of medical residents should be performed instead by medical teachers and other medical professionals. 

One of the participants stated, “*The rural hospital needs more comprehensive methods to educate the medical residents. A lack of work can disrupt education. Various professionals should be involved in education and share their challenges to improve educational systems*” (Participant 4, acute care ward).

## 4. Discussion

This study clarified nurses’ contributions to rural family medicine education. Quantitative analysis showed that their contributions in terms of teaching nursing and emotional support were low overall and lower among nurses working in acute care wards than among those working in chronic care wards, which may be affected by their busy working conditions. The qualitative analysis clarified that rural nurses’ contribution to rural family medicine education focused on education regarding professionalism, interprofessional collaboration, and respect for nurses’ working environment and competence. In addition, they struggle to educate medical residents on busy routine work; this education should be supplemented by various medical professionals.

Based on the quantitative results, rural nurses’ busy working conditions may affect their ideas of the roles of teaching nursing and providing emotional support in acute care wards. In this study, rural nurses perceived rural family medical education negatively regarding teaching nursing and providing emotional support to the residents, and the trend was stronger among nurses working in acute care wards. Working conditions can affect the quality of education due to the staff’s lack of time and mental stress [32,33]. Patient characteristics in acute care wards may affect rural nurses’ ideas regarding education [34]. For patients with acute care conditions, various medical professionals must promptly act during care [35,36]. In such situations, educational attitudes may be impeded, and specific educational skills should be taught [5,37]. However, there is a lack of education on how to teach medical residents in rural areas. Working stress and the characteristics of patients may affect rural nurses’ ideas of their role in education.

The qualitative analysis extracted three themes regarding rural nurses’ contributions to family medicine education: nurturing professionalism, driving interprofessional collaboration, and respect for nurses’ working environments and competencies. The first theme, nurturing professionalism, demonstrated the importance of nurses’ involvement in medical residents’ professional education. As this study shows, in the process of educating physicians, they are educated to be medical professionals and should be responsible for patient care. To become a medical professional, doctors need various clinical experiences based on knowledge from medical teachers [38,39]. Nurses can observe medical residents’ behaviors more frequently than other professionals and support their practices [40]. Nurses’ involvement in education regarding professionalism is essential in rural areas with a minimal workforce [41,42].

The second theme, driving interprofessional collaboration, suggests that effective education regarding ethical aspects can be facilitated by interprofessional collaboration in family medicine education. Older, frail patients have multiple biopsychosocial problems impinging on their quality of life [43,44]. Like this article and previous research show, medical residents struggle to manage their treatment and care [21,45]. Nurses and other medical professionals support treatment and care while respecting patients’ and families’ decisions [46,47]. In the process of interprofessional collaboration, medical residents can learn how to manage ethical issues, such as the value of extending lives and resuscitation among terminal and older patients [43,44]. Learning can be driven through dialogue and reflection with not only medical teachers but also nurses [48]. Interprofessional collaboration, including ethical issues, should be promoted in rural family medical education. Improved interprofessional collaboration could contribute to better-quality patient care.

The third theme, respect for nurses’ working environment and competence, indicates that effective family medicine education in rural areas requires nurses to educate medical residents regarding their working conditions and competence to reduce the burden on nurses. For such education, comprehensive approaches involving various professionals in the process of education, which can lead to better community care, can be essential. In this research, rural nurses struggled to educate medical residents in busy routine work, which should be communicated to medical residents for their effective work and education. The involvement of not only nurses and medical teachers but also social workers and therapists in educational processes can be effective if their specialties are respected [13,49,50]. In addition, other people in communities can be involved in education, which can be effective in rural medical education [51,52,53]. Their involvement can create opportunities to better understand the community, leading to respect for health conditions and, ultimately, to better healthcare in rural areas [51,52,53]. Future studies should investigate the quality of rural family medicine education by involving various medical professionals and people in rural communities.

The current study results have significant implications for future research. Accordingly, rural family medicine education should include clinical nurses for the effective provision of education, particularly regarding professionalism, interprofessional collaboration, and residents’ smooth transition to rural work conditions. Based on our research findings, clinical nurses could contribute to the professional education of medical residents through continual dialogue with them. Resident–nurse collaboration in clinical situations could drive the residents’ learning regarding interprofessional collaborations. Nurses and other professionals’ education related to rural hospitals’ work environments and competencies could mitigate conflicts among professionals.

This study had several limitations. First, as it was performed at a single rural Japanese community hospital, the results may have limited transferability. However, the educational system was described in depth, which may enable its application to other settings. The second limitation is confirmability. This study was conducted primarily by a medical educator in a community hospital, and the relationship between the interviewer and interviewees may have affected the content of the interviews. To mitigate these limitations, the third author discussed the contents with the second author, who was a nursing specialist. In addition, to improve confirmability and transferability, the authors discussed and reflected on the qualitative data, concepts, and themes that reached theoretical saturation.

## 5. Conclusions

This study clarified that rural nurses’ ideas of their role in family medicine education may be associated with their working conditions. Rural nurses’ education of family medicine residents focused on professionalism, interprofessional collaboration, and respect for working culture and competency. Rural nurses may perceive their role in such education as challenging. Rural family medicine education should incorporate clinical nurses as educators on professionalism and interprofessional collaboration. To this end, other professionals should be more actively involved in improving the quality of education.

## Figures and Tables

**Table 1 ijerph-19-03090-t001:** Participants’ demographics and the results of the questionnaire on nurses’ roles in family medicine education.

		Character of Care		
Factor	Total	Chronic Care	Acute Care	*p*-Value
N	88	43	45	
Clinical experience, average (SD)	20.16 (8.86)	21.42 (9.26)	18.96 (8.39)	0.194
Educational background (%)				
Specialized university, N (%)	17 (19.3)	7 (16.3)	10 (22.2)	0.163
General university, N (%)	8 (9.1)	2 (4.7)	6 (13.3)	
Specialized school, N (%)	63 (71.6)	34 (79.1)	29 (64.4)	
Role in resident education				
Guardians of patient wellbeing, average (SD)	3.93 (0.88)	3.93 (0.77)	3.93 (0.99)	0.987
Navigators, average (SD)	4.30 (0.76)	4.30 (0.71)	4.29 (0.82)	0.935
Providers of general support, average (SD)	4.11 (0.88)	4.05 (0.82)	4.18 (0.94)	0.486
Nurses as teachers, average (SD)	3.23 (1.07)	3.49 (0.67)	2.98 (1.31)	0.024
Providers of emotional support, average (SD)	2.69 (1.10)	2.93 (0.88)	2.47 (1.24)	0.047
Expert advisors, average (SD)	3.82 (0.86)	3.95 (0.69)	3.69 (1.00)	0.153
Team players, average (SD)	3.95 (0.83)	4.00 (0.72)	3.96 (0.82)	0.789

Notes: Guardians of patient well-being: Nurses must provide educational support to medical residents to safeguard patient safety; Navigators: Nurses must convey the background and concrete information of patients and their families to support medical care by residents; Providers of general support: Nurses must teach how to work in wards and how to prepare medical equipment so that residents can work smoothly in wards; Nurses as teachers: Nurses must support residents’ medical care to grow their personality and ability; Providers of emotional support: Nurses must play a supporting role in residents’ emotional changes; Expert advisors: Nurses must provide residents with knowledge of patient care as nursing specialists; Team players: Nurses must help residents make appropriate decisions in patient care; SD: Standard deviation.

**Table 2 ijerph-19-03090-t002:** Qualitative analysis of nurses’ role in family medicine education.

Theme	Concept
Nurturing professionalism	Responsibility as a physician
	Respecting patients’ backgrounds
	Enhancing ethical attitude
Driving interprofessional collaboration	Getting feedback from teachers and nurses
	Importance of dialogue with other professionals
	Quality improvement of care through collaboration
Respect for the environment and nurses’ competence	Understanding working environments and culture
	Working with respect for nurse’s competence

## Data Availability

The datasets used and/or analyzed during the current study may be obtained from the corresponding author upon reasonable request.

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
