# Peer review of "Nurses’ Contributions in Rural Family Medicine Education: A Mixed-Method Approach"

_ijerph, 2022, doi:10.3390/ijerph19053090_

Round 1

Reviewer 1 Report

The purpose of the study was to examine how nurses perceive their role in rural family medicine education.  The rationale for this study is that: (1) there is a lack of evidence regarding nurses’ perceptions of their involvement in rural family medical education, and (2) nurses have the potential to be valuable collaborators in family medical education since their involvement can be critical for the quality of care for patients. By clarifying these perceptions, education of family medicine residents in rural settings could be revised to enhance interprofessional collaboration in patient care education.

The authors conducted a mixed-method design for this study.  Eighty-eight nurse participants from a small rural hospital completed a questionnaire on their perceptions of their role in family medicine residents’ education.  One of the co-authors with expertise in family medicine, medical education, and public health prepared field notes from ethnographic observations of interactions between medical residents and nurses in various hospital wards and interviewed 20 nurses with administrative roles. Acute care nurses perceived lower levels of endorsement of their roles as “nurses as teachers” and “providers of emotional support” compared to chronic care nurses.  Qualitative analyses of the interviews yielded three main themes: personal growth, inter- and intra-professional collaboration, and respect for working culture.  The authors conclude that the study clarified rural nurses’ perceptions of their role in family medicine education.

This is primarily a qualitative study focused on nurses’ perceptions of their roles in educating medical residents. Given that little is known about such perceptions, the qualitative findings offer some insight into these perceptions and the challenges nurses in rural settings face.

Suggestions for further strengthening the manuscript are listed below:

  1. Page 2, lines 85-87. The authors indicate that in 2020, 39% of the total population in Unnan City were over 65 years old.  In the next line, they indicate that this percentage is expected to reach 50% by 2020 – the same year.  It would help if the authors could double-check the years and correct, if appropriate.
  2. Page 3, section 2.3. The section on participants should indicate how many nurses completed the questionnaire and how many were interviewed.  At present, the reader does not learn this information until the Results section. 
  3. Page 3, description of questionnaire. Consider replacing “questions” with “items” since they are worded as statements to be answered on a five-point Likert scale.
  4. There is an imbalance between the quantitative and qualitative aspects of this manuscript. More could be done to interpret the quantitative findings comparing the results for acute care versus chronic care nurses in Table 1 and at the bottom of page 4.  The authors report the findings, but there is no attempt to interpret the results.  Why do the acute care nurses report less endorsement of “nurses as teachers” and “providers of emotional support” compared to chronic care nurses?  Some explanation of such a finding, with support from any relevant literature, is warranted.
  5. Table 2 presents three main themes that emerged from the qualitative analysis:  (1) personal growth (subthemes are nurturing professionalism, respecting patients’ backgrounds, and enhancing ethical attitude), (2) Inter- and intra-professional collaboration (subthemes are collaboration between educators and learners, importance of dialogue with other professionals, and quality improvement of care through collaboration), and (3) Respect for working culture (subthemes are understanding working environments and culture, respecting care burden impinging on education).  However, throughout the rest of the manuscript, including in the abstract, the authors dwell on professionalism, ethical issues, and interprofessional collaboration, which are somewhat different from the three main themes displayed in Table 2.  It would help if the authors could clarify the dominant themes and present them in a more consistent manner. 
  6. The Conclusions section does not seem to be consistent with the extensive focus on themes shown in Table 2. Nor do the conclusions return to one of the rationales for conducting the study, i.e., that the findings should be helpful in providing guidance for revising education of family medicine residents in a way that enhances interprofessional collaboration in the patient care setting. Instead, the conclusions seem to dwell on the challenges nurses face and their working conditions. It would help if the authors provided a more thoughtful set of conclusions that correspond to the original rationale and that have implications for educating family medicine residents in rural settings, and ultimately to the quality of patient care. 

Author Response

Responses to the reviewers’ comments

Thank you very much for reviewing our manuscript and providing suggestions for its improvement. We have provided point-by-point responses to the reviewers’ comments; our revisions are indicated in red font in the manuscript. We hope that the revised manuscript meets the journal’s requirements and can now be considered for publication.

The purpose of the study was to examine how nurses perceive their role in rural family medicine education.  The rationale for this study is that: (1) there is a lack of evidence regarding nurses’ perceptions of their involvement in rural family medical education, and (2) nurses have the potential to be valuable collaborators in family medical education since their involvement can be critical for the quality of care for patients. By clarifying these perceptions, education of family medicine residents in rural settings could be revised to enhance interprofessional collaboration in patient care education.

Response:

Thank you for your valuable feedback.

The authors conducted a mixed-method design for this study.  Eighty-eight nurse participants from a small rural hospital completed a questionnaire on their perceptions of their role in family medicine residents’ education.  One of the co-authors with expertise in family medicine, medical education, and public health prepared field notes from ethnographic observations of interactions between medical residents and nurses in various hospital wards and interviewed 20 nurses with administrative roles. Acute care nurses perceived lower levels of endorsement of their roles as “nurses as teachers” and “providers of emotional support” compared to chronic care nurses.  Qualitative analyses of the interviews yielded three main themes: personal growth, inter- and intra-professional collaboration, and respect for working culture.  The authors conclude that the study clarified rural nurses’ perceptions of their role in family medicine education.

Response:

Thank you for your valuable feedback.

This is primarily a qualitative study focused on nurses’ perceptions of their roles in educating medical residents. Given that little is known about such perceptions, the qualitative findings offer some insight into these perceptions and the challenges nurses in rural settings face.

Response:

Thank you for your valuable feedback.

Suggestions for further strengthening the manuscript are listed below:

  1. Page 2, lines 85-87. The authors indicate that in 2020, 39% of the total population in Unnan City were over 65 years old.  In the next line, they indicate that this percentage is expected to reach 50% by 2020 – the same year.  It would help if the authors could double-check the years and correct, if appropriate.

Response:

Thank you for your valuable feedback. We have revised the relevant statement to include the year “2050,” as per the corresponding reference (lines 89-93).

“Unnan City is one of the smallest and most remote cities in Japan and is located southeast of an administrative unit in a rural setting. In 2020, the total population of the city was 37,638 (18,145 males and 19,492 females), and 39% were aged over 65 years; this statistic is expected to reach 50% by 2050. This city has 16 clinics, 12 home care stations, 3 visiting nurse stations, and only 1 public hospital.”

  1. Page 3, section 2.3. The section on participants should indicate how many nurses completed the questionnaire and how many were interviewed.  At present, the reader does not learn this information until the Results section. 

Response:

Thank you for your valuable feedback. In accordance with your comment, we have added the relevant information (lines 125–127).

“Overall, 88 nurses completed this distributed questionnaire, and of these, 20 nurses with administrative roles were interviewed based on the results of their questionnaires.”

  1. Page 3, description of questionnaire. Consider replacing “questions” with “items” since they are worded as statements to be answered on a five-point Likert scale.

Response:

Thank you for your valuable feedback. We have revised the term “question” to “item” throughout the corresponding text as per your recommendation (lines 130–147).

“A questionnaire was provided to the participants regarding their roles in family medicine residents’ education. Based on a previous study, seven items were constructed with respect to the concepts of previous research: nurses as teachers, guardians of patient well-being, providers of emotional support, providers of general support, expert advisors, navigators, and team players [15]. The seven items were as follows: nurses need to provide educational support to medical residents to protect the safety of patients (Item 1: guardians of patient well-being); nurses need to convey the background and concrete information of patients and their families to support the medical care of residents (Item 2: navigators); nurses need to teach how nurses work in wards and how to prepare medical equipment so that residents can work smoothly in wards (Item 3: providers of general support); nurses need to support the medical care of residents to grow their personality and ability (Item 4: nurses as teachers); nurses need to play a supporting role in the emotional changes of residents (Item 5: providers of emotional support); nurses need to provide residents with knowledge regarding patient care as nursing specialists (Item 6: expert advisors); and nurses need to help residents make appropriate decisions in patient care (Item 7: team players). Each item was answered on a five-point Likert scale ranging from strongly agree (five) to strongly disagree (one). In addition, the gender, clinical experience, workplace, and educational background of the participants were collected.”

  1. There is an imbalance between the quantitative and qualitative aspects of this manuscript. More could be done to interpret the quantitative findings comparing the results for acute care versus chronic care nurses in Table 1 and at the bottom of page 4.  The authors report the findings, but there is no attempt to interpret the results.  Why do the acute care nurses report less endorsement of “nurses as teachers” and “providers of emotional support” compared to chronic care nurses?  Some explanation of such a finding, with support from any relevant literature, is warranted.

Response:

Thank you for your valuable feedback. As per your comment, we have added further details regarding the quantitative results above Table 1 (lines 199–207).

“The nurses’ average clinical experience was 20.16 years (standard deviation [SD] = 8.86), and most participants graduated from specialized nursing schools. All of the participants were women. Regarding the questionnaire, the scores for items on “nurses as teachers” and “providers of emotional support” were statistically lower among the participants working in acute care wards than those working in chronic care wards (p = 0.024 and 0.047, respectively). The other items regarding nurses’ roles—“guardians of patient wellbeing,” “navigators,” “providers of general support,” “expert advisors,” and “team players”—had higher scores than the roles of “nurses as teachers” and “providers of emotional support” but did not report any significant statistical differences.”

  1. Table 2 presents three main themes that emerged from the qualitative analysis:  (1) personal growth (subthemes are nurturing professionalism, respecting patients’ backgrounds, and enhancing ethical attitude), (2) Inter- and intra-professional collaboration (subthemes are collaboration between educators and learners, importance of dialogue with other professionals, and quality improvement of care through collaboration), and (3) Respect for working culture (subthemes are understanding working environments and culture, respecting care burden impinging on education).  However, throughout the rest of the manuscript, including in the abstract, the authors dwell on professionalism, ethical issues, and interprofessional collaboration, which are somewhat different from the three main themes displayed in Table 2.  It would help if the authors could clarify the dominant themes and present them in a more consistent manner. 

Response:

Thank you for your valuable feedback. In accordance with your suggestion, we have revised the manuscript to include consistent terminology and details regarding the themes and concepts that emerged via the qualitative analysis throughout the manuscript.

  1. The Conclusions section does not seem to be consistent with the extensive focus on themes shown in Table 2. Nor do the conclusions return to one of the rationales for conducting the study, i.e., that the findings should be helpful in providing guidance for revising education of family medicine residents in a way that enhances interprofessional collaboration in the patient care setting. Instead, the conclusions seem to dwell on the challenges nurses face and their working conditions. It would help if the authors provided a more thoughtful set of conclusions that correspond to the original rationale and that have implications for educating family medicine residents in rural settings, and ultimately to the quality of patient care. 

Response:

Thank you for your valuable feedback. We have revised the Conclusion section to be more consistent with the rationale and research question (Line 498-504).

“This study clarified that rural nurses’ ideas of their role in family medicine education may be associated with their working conditions. Rural nurses’ education of family medicine residents focused on professionalism, interprofessional collaboration, and respect for working culture and competency. Rural nurses may perceive their role in such education as challenging. Rural family medicine education should incorporate clinical nurses as educators on professionalism and interprofessional collaboration. To this end, other professionals should be more actively involved to improve the quality of education.”

Reviewer 2 Report

The manuscript reports about a mixed method study about the "perception" of nurses in a rural city hospital on their role in family medicine education.

The topic is of interest and the Introduction shows that research lacks in this particular field, while the role of nurses in residents' education has been already discussed for other contexts.

My main concerns are about the  research question, the method and the way results are presented and discussed

Research question

It is described as "how do nurses perceive the difficulties and provision of support in rural family medicine education?" (page 2, lines 67-68). A first comment is about the word "perception", which usually means both "awareness and interpretation of a physical sensation or lived experience". If so, the best method to explore perception is phenomenology. Maybe better wordings are: opinions ... believes ... ideas ... conceptions ...
The questionnaire explores a set of possible educational roles of nurses, the interview has a first question about "feelings" (How do you feel about working with family medicine residents?) but then go on asking questions about opinions and the results have been interpreted identifying nurses' opinions about what the residents would need to develop a mature professionalism and clinical competence in such a setting as a rural hospital and how the nurses could help this process.

Method: the authors adopted a mixed method approach, with a quantitative survey derived from a previous study and an etnography analized according to grounded theory. Mixed methods are useful if the researcher clarify how the results of each method enlights the interpretation of the results of the other method or drive the other method itself (definition of questions, choose of the approach, ...).
The definition of the survey is well done, based on the prevoius study, but then the results of the survey are completely out of context, just put beside the results of the qualitative study. There are no cross references nor in term of confirmation nor in term of contraddiction.

Grounded theory is meant to develop a "theory", and the authors state that "Finally, the theory was discussed by two authors" (page 4, lines 170-71). Unfortunately, neither the results nor the discussion reports  a "theory", that is to say a system of concepts and/or principles, linked in a logical and coherent way, useful to explain facts or events. The simple listing of themes is not a theory.
Hence the authors should organize their findings in a coherent view, that can be used - for example - to design or to interpret an  educational environment.

Results

The labels of the themes are unclear. Often the subject of the theme is missing, like in "personal growth", which I translated in "Residents' needs for education and personal growth" or "Nursing professionalism", that is about residents' professionalism!! "Collaboration Between Teachers and Residents" is more likely "Get feedback from teachers and residents".
Finally, I'm not sure I could understand "Respect for Working Culture". It could be "Knowledge and respect for the environment and for nurses' competence", but I'm not sure.

Overall, the authors should decide the point of view: are you listing the "educational needs nurses detected" or - as stated - "nurses’ perceptions of their roles in rural family medicine education in rural community hospitals"?That is to say: are you focussing nurses' analysis of needs or their role as educational agents in that context or their lived experience, as when they declare that they would like to be more respected and listened? 
Unfortunately, I found the results (and hence the discussion) messy and confusing.

Discussion

The general strucuture of a Discussion should be:
Main findings, our findings are coherent with ... and add new knowledge on ... The implications for implementation and further research and are ... limitations

More or less, the reader can find these sections in the text, but the sections could be better organized, making the content explicit and clearly addresing the discussion to the research question(s) and to the context of the study.

I suggest a deep revision of the whole study, accoridng to the suggestions I tried to give, because the study is of interest and I'm sure the authors collected valuable results: just make clear to the reader what you meant to show and be coherent with the adopted methods.

Minor issues

Page 2, lines 56-7 "Rural family medicine residents experience a wide scope of practice as a result of the needs of rural hospitals; therefore, nurses’ support and feedback can be more valuable [20]." I understand that reference #20 gives the needed explanation, but the reader would be happy to know here a little bit more on WHY "therefore, nurses’ support and feedback can be more valuable"

Page 2, lines 63-4: "rural family medicine education should be provided effectively to sustain interprofessional collaboration" I'd say that interprofessional collaboration sustains a good medical education ...

Page 2, lines 86-7: "this is expected to reach 50% by 2020" . Update the sentence, in the meantime we are in 2022...

Page 3, line 109: "In the first year, which began on April 1," First year of what?

Page 4, line 154: chi-square test. You only compared mean scores, probably you only used t test (if the series were normally distributed...)

Page 9, line 385: "the contents of teaching nursing". Do nurses "teach" contents, like in lectures?

Conclusions: it is only a repetition of what already written. Try to elaborate a "take home message" and opening to the future

Author Response

Responses to the reviewers’ comments

Thank you very much for reviewing our manuscript and providing suggestions for its improvement. We have provided point-by-point responses to the reviewers’ comments; our revisions are indicated in red font in the manuscript. We hope that the revised manuscript meets the journal’s requirements and can now be considered for publication.

The manuscript reports about a mixed method study about the "perception" of nurses in a rural city hospital on their role in family medicine education.

The topic is of interest and the Introduction shows that research lacks in this particular field, while the role of nurses in residents' education has been already discussed for other contexts.

My main concerns are about the  research question, the method and the way results are presented and discussed

Research question

It is described as "how do nurses perceive the difficulties and provision of support in rural family medicine education?" (page 2, lines 67-68). A first comment is about the word "perception", which usually means both "awareness and interpretation of a physical sensation or lived experience". If so, the best method to explore perception is phenomenology. Maybe better wordings are: opinions ... believes ... ideas ... conceptions ...

Response:

Thank you for your valuable feedback. Accordingly, we have revised the word “perception” to “ideas” throughout the manuscript.

The questionnaire explores a set of possible educational roles of nurses, the interview has a first question about "feelings" (How do you feel about working with family medicine residents?) but then go on asking questions about opinions and the results have been interpreted identifying nurses' opinions about what the residents would need to develop a mature professionalism and clinical competence in such a setting as a rural hospital and how the nurses could help this process.

Method: the authors adopted a mixed method approach, with a quantitative survey derived from a previous study and an etnography analized according to grounded theory. Mixed methods are useful if the researcher clarify how the results of each method enlights the interpretation of the results of the other method or drive the other method itself (definition of questions, choose of the approach, ...).

The definition of the survey is well done, based on the prevoius study, but then the results of the survey are completely out of context, just put beside the results of the qualitative study. There are no cross references nor in term of confirmation nor in term of contraddiction.

Response:

Thank you for your valuable feedback. Following your comments, we have revised the Material and Methods section to provide further details regarding the connection between qualitative and quantitative data by elaborating upon the use of quantitative data for the qualitative data inquiry (lines 156–167).

“The first question was “How do you feel about the current family medicine education in community hospitals?”

- The follow-up questions focused on the accomplishing of their education of family medicine residents.

The second question was “What do you think you can do in family medicine education at a community hospital?”

- The follow-up questions focused on how the nurses educated family medicine residents as per the positive quantitative results for the questionnaire items.

The third question was “How do you think your role impedes family medicine education at a community hospital?”

- The follow-up questions focused how the nurses educated family medicine residents as per the negative quantitative results for the questionnaire items.”

Grounded theory is meant to develop a "theory", and the authors state that "Finally, the theory was discussed by two authors" (page 4, lines 170-71). Unfortunately, neither the results nor the discussion reports  a "theory", that is to say a system of concepts and/or principles, linked in a logical and coherent way, useful to explain facts or events. The simple listing of themes is not a theory.

Hence the authors should organize their findings in a coherent view, that can be used - for example - to design or to interpret an  educational environment.

Response:

Thank you for your valuable feedback. We have comprehensively revised the manuscript and focused on nurses’ contributions and difficulties in rural family medicine education as per your comment. The following parts are the main revisions.

Line 56-76

“Rural family medicine residents experience a wide scope of practice owing to their demand in rural hospitals; therefore, nurses’ support and feedback can be more valuable in rural family medicine education [20]. Rural family medicine education may involve various conflicts due to systemic and cultural changes for medical residents, as they may have to change their working styles in adjusting to rural clinical situations [21–23]. In these processes, as the number of senior doctors is low, nurses may play critical roles in supporting residents’ conflicts [24]. As nurses frequently observe the residents, they could provide residents with various educational recommendations to improve their collaboration. Moreover, effective support and safe netting for patient care should be provided. Furthermore, in rural areas, there is a lack of physicians; therefore, interprofessional collaboration facilitates a good medical education in rural family medicine [25,26].

Currently, there is a lack of evidence regarding nurses’ contributions to rural family medicine education [15]. In addition, nurses’ difficulties in rural family medicine education have not been clarified. Therefore, the research question was “How do nurses contribute to rural family medicine education, and what difficulties do they experience?” By clarifying nurses’ contribution and difficulties in rural family medicine education, a concrete revision of such education can be executed, which may lead to better interprofessional collaboration in patient care education. Therefore, this research aimed to clarify nurses’ contributions and difficulties in rural family medicine education using a mixed-method approach.”

Results

The labels of the themes are unclear. Often the subject of the theme is missing, like in "personal growth", which I translated in "Residents' needs for education and personal growth" or "Nursing professionalism", that is about residents' professionalism!! "Collaboration Between Teachers and Residents" is more likely "Get feedback from teachers and residents".

Finally, I'm not sure I could understand "Respect for Working Culture". It could be "Knowledge and respect for the environment and for nurses' competence", but I'm not sure.

Response:

Thank you for your valuable feedback. We have thoroughly revised the themes and concepts in the Results section according to the research rationale and question. The following parts are the main revisions.

Line 227-237

“The nurses interacted with the family medicine residents and realized the residents should improve their quality of professionalism while caring for patients as medical doctors. They attempted to discuss these behaviors with the residents as authentic physicians. Residents tended to decide on various treatments and care based primarily on medical aspects. However, the nurses attempted to incorporate into residents’ decision-making respect for patients’ backgrounds to facilitate effective care because various ethical decisions are made in geriatric medicine. When the residents struggled with ethical decision making, the nurses conversed with the residents to support their discussions with patients and their families to effectively address these issues. Consistent with our quantitative results, nurses functioned as guardians of patient well-being, navigators, and expert advisors.”

Line 239-247

“The nurses realized that residents needed to modify their attitudes toward patients, families, and other medical staff as professional physicians, which could be supported by the nurses. Family medicine residents were trained in various medical situations by their teachers but had no experience of authentic responsibility for patient care in previous situations. In the rural hospital, they had to determine patients’ treatment and care plans in outpatient and inpatient departments. In these processes, the residents’ ambiguous attitudes and vague decision making confused the nurses. Nurses tried to modify residents’ attitudes toward medical care. They realized that before determining medical care, medical residents should nurture themselves as authentic physicians.”

Line 262-269

“The nurses observed that the medical residents did not respect patients’ backgrounds and needed to include psychosocial aspects in their decision making. Residents’ decision making primarily focused on biomedical conditions and did not consider patients’ lives in their homes or nursing homes. The nurses knew that the medical residents learned the biopsychosocial model, in which family physicians understand patients from not only a biomedical perspective but also a psychosocial perspective to provide better care. However, the residents’ skills and attitudes needed to be enhanced through clinical experience and collaboration with nurses.”

Line 285 – 290

“Through residents’ improvement as authentic physicians and respecting patients’ background, the nurses believed that medical residents could learn about ethical attitudes in the treatment of older and frail patients through discussion with nurses. Medical residents are exposed to various ethical issues when caring for older patients. The residents struggled with ethical decision making because of their lack of experience. The nurses realized that they could effectively support the residents through dialogue regarding patients’ quality of life.”

Line 304-312

“Nurses emphasized the importance of collaboration. They believed that medical teachers as well as nurses should first give detailed feedback to the residents regarding effective collaboration. In clinical situations, the nurses educated residents regarding the collaboration among medical professionals. In addition, nurses considered that through various educational insights and experiences, the medical residents could realize the effectiveness of interprofessional collaboration in improving patient care. Thus, consistent with the quantitative results, nurses functioned as providers of general support and team players.”

Overall, the authors should decide the point of view: are you listing the "educational needs nurses detected" or - as stated - "nurses’ perceptions of their roles in rural family medicine education in rural community hospitals"?That is to say: are you focussing nurses' analysis of needs or their role as educational agents in that context or their lived experience, as when they declare that they would like to be more respected and listened? 

Unfortunately, I found the results (and hence the discussion) messy and confusing.

Response:

Thank you for your valuable feedback. In accordance with your comment, we have signficantly revised the Results and Discussion sections to focus on nurses’ contributions to rural family medicine education.

Discussion

The general strucuture of a Discussion should be:

Main findings, our findings are coherent with ... and add new knowledge on ... The implications for implementation and further research and are ... limitations

Response:

Thank you for your valuable feedback. Accordingly, we have thoroughly revised the Discussion section by including an explanation of three themes and the implications of the current study findings for future research.

More or less, the reader can find these sections in the text, but the sections could be better organized, making the content explicit and clearly addresing the discussion to the research question(s) and to the context of the study.

Response:

Thank you for your valuable feedback. As per your recommendations, we have revised and restructured the contents of the Results, Discussion, and Conclusion sections comprehensively to be consistent with the research questions.

I suggest a deep revision of the whole study, accoridng to the suggestions I tried to give, because the study is of interest and I'm sure the authors collected valuable results: just make clear to the reader what you meant to show and be coherent with the adopted methods.

 Response:

Thank you for your valuable feedback. Following your detailed comments, we have significantly revised this entire paper to improve the consistency and readability of the text. Please review our revisions and consider our manuscript for publication.

Minor issues

Page 2, lines 56-7 "Rural family medicine residents experience a wide scope of practice as a result of the needs of rural hospitals; therefore, nurses’ support and feedback can be more valuable [20]." I understand that reference # 20 gives the needed explanation, but the reader would be happy to know here a little bit more on WHY "therefore, nurses’ support and feedback can be more valuable"

Response:

Thank you for your valuable feedback. Accordingly, we have revised the relevant text by adding an explanation regarding the relevance of the reference to the current study (lines56–67).

“Rural family medicine residents experience a wide scope of practice owing to their demand in rural hospitals; therefore, nurses’ support and feedback can be more valuable in rural family medicine education [20]. Rural family medicine education may involve various conflicts due to systemic and cultural changes for medical residents, as they may have to change their working styles in adjusting to rural clinical situations [21–23].”

Page 2, lines 63-4: "rural family medicine education should be provided effectively to sustain interprofessional collaboration" I'd say that interprofessional collaboration sustains a good medical education ...

Response:

Thank you for your valuable feedback. Per your comment, we have revised the suggested portion of text (lines 56–67).

“In these processes, as the number of senior doctors is low, nurses may play critical roles in supporting residents’ conflicts [24]. As nurses frequently observe the residents, they could provide residents with various educational recommendations to improve their collaboration. Moreover, effective support and safe netting for patient care should be provided. Furthermore, in rural areas, there is a lack of physicians; therefore, interprofessional collaboration facilitates a good medical education in rural family medicine [25,26].”

Page 2, lines 86-7: "this is expected to reach 50% by 2020" . Update the sentence, in the meantime we are in 2022...

Response:

Thank you for your valuable feedback. We have revised the relevant statement to include the year “2050,” as per the corresponding reference (lines 89-93).

“Unnan City is one of the smallest and most remote cities in Japan and is located southeast of an administrative unit in a rural setting. In 2020, the total population of the city was 37,638 (18,145 males and 19,492 females), and 39% were aged over 65 years; this statistic is expected to reach 50% by 2050. This city has 16 clinics, 12 home care stations, 3 visiting nurse stations, and only 1 public hospital.”

Page 3, line 109: "In the first year, which began on April 1," First year of what?

Response:

Thank you for your valuable feedback. We have revised the corresponding part as per your recommendation (lines 114–117).

“In the first year of the training, which began on April 1, medical residents collaborated with various medical professionals at the community hospital. This curriculum can be utilized to educate a maximum of three residents simultaneously. One resident in 2018 and 2019 and three in 2020 and 2021 engaged in the curriculum.”

Page 4, line 154: chi-square test. You only compared mean scores, probably you only used t test (if the series were normally distributed...)

Response:

Thank you for your valuable feedback. We have revised the corresponding text as per your recommendation (lines 172–1177).

“Quantitative data were analyzed using Student’s t-test and a chi-square test for the background data. The results of each question regarding nurses’ roles in family medicine education were compared between the characteristics of the wards in which the participants worked: acute or chronic care using Student’s t-test. Regarding qualitative data, the grounded theory approach was used to clarify nurses’ contributions and difficulties in regard to rural family medicine education in rural community hospitals.”

Page 9, line 385: "the contents of teaching nursing". Do nurses "teach" contents, like in lectures?

Response:

Thank you for your valuable feedback. We have revised the corresponding text as per your recommendation.

Conclusions: it is only a repetition of what already written. Try to elaborate a "take home message" and opening to the future

Response:

Thank you for your valuable feedback. We have revised the Conclusion section to be more consistent with the rationale and research question (Line 498-504).

“This study clarified that rural nurses’ ideas of their role in family medicine education may be associated with their working conditions. Rural nurses’ education of family medicine residents focused on professionalism, interprofessional collaboration, and respect for working culture and competency. Rural nurses may perceive their role in such education as challenging. Rural family medicine education should incorporate clinical nurses as educators on professionalism and interprofessional collaboration. To this end, other professionals should be more actively involved to improve the quality of education.”

Round 2

Reviewer 2 Report

I'm satisfied with the changes the authors made to the manuscript, which is now clearer and fully provide its contribution to the advancement of knowledge in this specific field